# Changes in the Transmission Dynamic of Chikungunya Virus in Southeastern Senegal

**DOI:** 10.3390/v12020196

**Published:** 2020-02-10

**Authors:** Abdourahmane Sow, Birgit Nikolay, Oumar Faye, Simon Cauchemez, Jorge Cano, Mawlouth Diallo, Ousmane Faye, Bakary Sadio, Oumar Ndiaye, Scott C. Weaver, Anta T. Dia, Amadou Alpha Sall, Denis Malvy

**Affiliations:** 1Institut Pasteur Dakar, Arbovirus and viral Hemorrhagic Fevers Unit, 12500 Dakar, Senegal; Oumar.faye@pasteur.sn (O.F.); Ousmane.Faye@pasteur.sn (O.F.); Bakary.Sadio@pasteur.sn (B.S.); Oumar.Ndiaye@pasteur.sn (O.N.); Amadou.Sall@pasteur.sn (A.A.S.); 2Institut Santé et développement (ISED), Université Cheikh Anta Diop, 12500 Dakar, Senegal; diagodia@hotmail.fr; 3Inserm U1219 University of Bordeaux, 33063 Bordeaux, France; denis.malvy@chu-bordeaux.fr; 4Mathematical Modelling of Infectious Diseases Unit, Institut Pasteur, 75015 Paris, France; birgit.nikolay@pasteur.fr (B.N.); simon.cauchemez@pasteur.fr (S.C.); 5Centre National de Recherche Scientifique (CNRS), URA3012, Paris, France; 6Center of Bioinformatics, Biostatistics and Integrative Biology, Institut Pasteur, 75015 Paris, France; 7Faculty of Infectious Diseases and Tropical Medicine, London School of Hygiene & Tropical Medicine, London WC1E7HT, UK; jcano.ortega@lshtm.ac.uk; 8Institut Pasteur Dakar, Medical Entomology Unit, 12500 Dakar, Senegal; Mawlouth.Diallo@pasteur.sn; 9Institute for Human Infections and Immunity, World Reference Center for Emerging Viruses and Arboviruses and Department of Microbiology and Immunology, University of Texas Medical Branch, Galveston, TX 77555, USA; sweaver@utmb.edu

**Keywords:** Chikungunya, spatial autocorrelation, environmental risk, gold mining, Senegal

## Abstract

In Senegal, chikungunya virus (CHIKV) is maintained in a sylvatic cycle and causes sporadic cases or small outbreaks in rural areas. However, little is known about the influence of the environment on its transmission. To address the question, 120 villages were randomly selected in the Kedougou region of southeastern Senegal. In each selected village, 10 persons by randomly selected household were sampled and tested for specific anti-CHIKV IgG antibodies by ELISA. We investigated the association of CHIKV seroprevalence with environmental variables using logistic regression analysis and the spatial correlation of village seroprevalence based on semivariogram analysis. Fifty-four percent (51%–57%) of individuals sampled during the survey tested positive for CHIKV-specific IgG. CHIKV seroprevalence was significantly higher in populations living close to forested areas (Normalized Difference Vegetation Index (NDVI), Odds Ratio (OR) = 1.90 (1.42–2.57)), and was negatively associated with population density (OR = 0.76 (0.69–0.84)). In contrast, in gold mining sites where population density was >400 people per km^2^, seroprevalence peaked significantly among adults (46% (27%–67%)) compared to all other individuals (20% (12%–31%)). However, traditional gold mining activities significantly modify the transmission dynamic of CHIKV, leading to a potential increase of the risk of human exposition in the region.

## 1. Introduction

Chikungunya virus (CHIKV) is a mosquito–borne alphavirus that belongs to the *Togaviridae* family [1]. It was first isolated in 1953 from the serum of a febrile patient during an epidemic in the Newala district of Tanzania [2,3]. Acute CHIKV infection in humans can cause a flu-like syndrome associated with a rash and severe arthralgia [4,5,6,7,8]. In sub-Saharan Africa, CHIKV is maintained in a sylvatic cycle involving non-human primates as reservoir hosts [9,10] and forest-dwelling mosquitoes [11,12]. Sylvatic vectors can be responsible for sporadic cases or small outbreaks among humans living in rural areas [9,13,14]. In urban areas, CHIKV is transmitted among humans by *Aedes aegypti* and *Aedes albopictus* mosquitoes [15]. 

Since the outbreak in Tanzania in 1952, CHIKV outbreaks have been reported in Africa, Asia, and Latin America between the 1960s and 2000s [16]. Recently, CHIKV was recognized as an emerging arbovirus with an important public health impact after major epidemics occurred in 2004 in several countries (Kenya, Comores, and islands in the Indian Ocean). The largest outbreak in the Indian Ocean basin occurred in La Reunion Island in 2005, with 300,000 infected cases, and an attack rate of about 35% [17]. In addition to major epidemics in Asia from 2005 to 2008, a significant outbreak occurred in Italy in 2007, and sporadic autochthonous cases have been detected in northeastern Italy and southeastern France, as well as in the USA among travelers from India in 2006 [18,19,20,21]. The spread of CHIKV in Europe and parts of Asia was attributed to the spread of the anthropophilic mosquito *Ae. albopictus* outside of Asia via international transportation, and the global movement of viremic individuals [22,23,24], emphasizing CHIKV as a re-emerging threat to global public health. 

In Senegal, CHIKV was first isolated from a bat in 1962 [25,26], and since then, sporadic human cases and outbreaks were regularly reported [14,19,27,28,29]. Since 1972, the Pasteur Institute of Dakar has implemented an entomological surveillance program in the Kedougou area, located on the border of Guinea, in southeastern Senegal. In this area, CHIKV was repeatedly isolated from *Aedes furcifer, Aedes luteocephalus*, and Aedes *taylori* [9,29,30,31,32]. Amplifications of CHIKV have been detected at approximately 5-year intervals. This interval is hypothesized to be the time necessary for the turnover of susceptible vertebrate hosts, mainly nonhuman primates [6]. In 2009, a CHIKV zoonotic amplification occurred in the Kedougou region, detected in both humans and mosquitoes. Indeed, 20 confirmed human cases were reported in the Kedougou and Saraya districts, mainly in gold mining sites. In parallel, 42 CHIKV-infected mosquito pools were obtained by RT-PCR from September to December 2009, mainly from *Ae. furcifer* (16 pools), *Ae. taylori* (5 pools)*, and Ae. luteocephalus* (5 pools) [33].

Despite active sylvatic circulation of CHIKV in the Kedougou region, limited information is available about its impact on human health and its interaction with environmental conditions. To address these questions, we conducted a serosurvey in 2012 following the last detected virus amplification in 2009. Here, we report the results of the serosurvey implemented in Kedougou, southeastern Senegal.

## 2. Materials and Methods

### 2.1. Serological Study

The study was carried out in the Kedougou region, located in the extreme southeast of Senegal, between 12°33′ north latitude and 12°11′ west longitude (Figure 1A,B). It extends over an area of 16,896 km^2^ with an estimated population of 153,476 inhabitants, of which 55% are under 20 years of age, with an average density of 8 persons per km^2^ [34]. The population is predominantly rural (84%), and ethnically diverse. On average, annual rainfall in the area is estimated between 1200 and 1300 mm. Agriculture remains the principal economic activity, but traditional gold mining has increased considerably, leading to massive human migration and important eco-environmental changes.

The sampling method was based on a two-level cross-sectional randomized cluster sampling adapted from the WHO procedure. The sampling frame was the list of villages drawn up for the 2002 national census. The Kedougou region was first divided into 3 districts. For each district, 40 villages were randomly selected using the cumulative total method. In each of the selected villages, 10 persons by randomly selected household were sampled. From each consented individual, 5 mL of intravenous blood was taken. Samples were centrifuged, and serum aliquoted, and sent in liquid nitrogen to the Dakar Pasteur Institute for anti-CHIKV IgG antibodies ELISA, according to the method described by Traore-Lamizana et al. [35]. Briefly, indirect IgG ELISA testing was performed on Microplate ELISA 96-well Maxisorp. After a coating step with CHIKV mouse hyperimmune ascitic fluid, followed by CHIKV antigen capture, samples were added, and specific antibody-antigen complexes were revealed by an anti-human IgG horseradish peroxidase conjugated antibody (KPL, USA). An ELISA microplate reader showed the Optical Density (OD) and sera as positive samples if the OD was ≥0.20 and the ratio (R) between the sample and the negative control was >2.

### 2.2. Environmental Data

A suite of environmental, topographical, and demographic datasets was used to explore potential drivers of CHIKV outbreaks in the study area. From the Moderate Resolution Imaging Spectroradiometer (MODIS) [36] product, we downloaded global MOD13Q1 data, which include vegetation indices such as the Normalized Difference Vegetation Index (NDVI), the Enhanced Vegetation Index (EVI), and the mid-infrared band (MIR). The NDVI and EVI are effective for quantifying green vegetation but the EVI was specifically developed to be more sensitive to changes in areas having high biomass, while MIR has been found to be useful to discriminate water surfaces [37]. Forest cover for the study area was obtained from the Global Forest Change project (University of Maryland, Maryland, USA) [38]. The elevation dataset at 250 m resolution was derived from a gridded digital elevation model produced by the Shuttle Radar Topography Mission (SRTM) [39]. Finally, gridded maps at 100 m resolution of estimated population density for Senegal, in 2010 and 2015, were obtained from the World Pop project [39]. Environmental, topographic, and demographic data were extracted for village point locations as average values over a buffer zone of 1 km radius. We further assessed sensitivity of estimates to buffer size by repeating the analysis with a buffer zone of a 3 km radius to account for movement of individuals around village locations.

### 2.3. Statistical Analysis

#### 2.3.1. Descriptive Analysis

Age was classified as <5, 5 to 9, 10 to 19, 20 to 39, 40 to 59, and ≥60 years. Associations of seroprevalence with the age and sex of individuals were investigated by logistic regression analysis, and statistical significance was assessed with a likelihood ratio test. The models included random intercepts for villages and rural communities to adjust for clustering of surveyed individuals. Confidence intervals of seroprevalence by sex or age group were obtained based on the exact binomial method. We included a sex-age interaction term to explore potential greater exposure to infections on certain groups (i.e., male adults be more exposed during their activities outside their resident villages), considering that age groups reflect different occupational activities (<20, 20 to 59, and ≥60 years), and stratified by population density (locations with ≤400 and >400 people per km^2^).

#### 2.3.2. Spatial Patterns of Seroprevalence

We aggregated individuals by villages and rural communities to assess the spatial variation in seroprevalence levels. We investigated the spatial correlation of village seroprevalence based on semivariogram analysis using the geoR package.

#### 2.3.3. Environmental Risk Factors

We first investigated the association of CHIKV seroprevalence with environmental variables by univariable logistic regression analysis, including random intercepts for villages and rural communities. We classified environmental variables into quintiles to assess departure from linearity in associations and included these as categorical terms in the models. For each variable, we compared the model fit to a model including the variable as a continuous term. The decision to include variables as categorical or continuous terms was based on the lowest Akaike information criterion (AIC). For variables associated with seroprevalence that were highly correlated (Pearson’s *r* > 0.7), we performed a preliminary variable selection based on the lowest likelihood ratio test (LRT) *p*-values and lowest AIC. Due to convergence problems when including all variables simultaneously, we chose a forward model selection approach, starting with the variable with the lowest *p*-value and lowest AIC, adding additional variables in order of increasing *p*-values and AIC. Variables were retained in the model if significantly associated (*p* ≤ 0.05). Random intercepts were retained in the final model if these were significantly associated (*p* ≤ 0.05) and improved the model fit.

To assess spatial patterns in the unexplained variation of seroprevalence by villages, we investigated spatial correlation of village random-effects by semivariogram analysis, as described above. Additionally, we compared model fit of the selected logistic regression model to a geostatistical model that additionally accounted for spatial correlation between village random effects.

Basic statistical analysis was performed using the R computing environment, and parameters of geostatistical models were estimated using Bayesian methods implemented in Winbugs [40].

## 3. Results

### 3.1. Serological Investigation for CHIKV IgG in 2012

In total, 998 individuals living in 101 villages and 15 rural communities in the Kedougou region were tested for anti-CHIKV IgG. The age of tested individuals ranged from 1 to 99 years (median 21, IQR 12–41) and 56% of tested individuals were male. Fifty-four percent of individuals (51%–57%) were positive for CHIKV IgG. Seroprevalence did not vary significantly by sex (males 53% (95% confidence values 49–57); females 55% (50%–60%); *p* = 0.522) or age-group (*p* = 0.485) (Figure 2).

### 3.2. Spatial Variation in CHIKV IgG Seroprevalence

Seroprevalence against CHIKV varied among villages and rural communities in the study area (Figure 1A); however, village seroprevalence levels were not spatially correlated (Appendix A, Figure A1). Table 1 and Table A1 in Appendix B show that in univariate analysis, CHIKV seroprevalence was significantly higher in populations living in areas close to a forest with a large amount of vegetation, compared to those living in areas with less or without vegetation (NDVI, Odds Ratio (OR) = 1.90 (1.42–2.57)).

The model that best explained the observed spatial variation in seroprevalence was based on population density (Figure 1B) and accounted for clustering of individuals in villages (village random effects). Indeed, seroprevalence was negatively associated with population density, so that for each one-unit increase in population density at the log-scale, the seroprevalence decreased by an Odds Ratio (OR) of 0.76 (0.69–0.84) (Table 2 and Table A2).

This translates, for example, into a predicted seroprevalence of 57% (55%–62%) at a population density of 10 persons per km^2^, compared to 32% (23%–38%) at a population density of 500 persons per km^2^ (Figure 1C). Village random effects were not spatially correlated, and including spatial dependency did not improve model fit (Appendix A, Figure A1).

Contrary to Figure 1D, among individuals living in villages where population density was >400 people per km^2^ (i.e., seven villages in the rural community of Bandafassi, which are the main sites of traditional gold mining), seroprevalence against CHIKV peaked among adults; in particular, it was significantly higher among male adults 20–59 years old at 46% (27%–67%), compared to all other individuals (20% (12%–31%); *p* = 0.013) ( Table A3). There was suggestive evidence for an interaction between sex and age (p_interaction_ = 0.098).

Among individuals living at population densities ≤400 people per km^2^, seroprevalence among male adults did not differ significantly from other individuals (*p* = 0.091) and no interaction between age and sex was detected (p_interaction_ = 0.766).

## 4. Discussion

Two years after a chikungunya outbreak in the Kedougou region [41], our survey showed that over 50% of studied individuals had a history of CHIKV infection. Seroprevalence was homogeneously distributed over all age ranges, including very young children, suggesting a simultaneous and recent exposure of the population to CHIKV circulation. Distinctly, continuous endemic circulation of CHIKV within this population would have led to a significant age pattern with increasing seroprevalence by age.

Seroprevalence against CHIKV was highest in remote areas with low population density. Indeed, individuals living in those areas were 1.24 times more likely exposed to CHIKV than those living in areas with a high population density. This can be explained by CHIKV transmission through spillover via sylvatic mosquitoes, such as *Ae. furcifer*, which is more frequent in rural areas close to the forest galleries, and which was identified as the main vector during the 2009 epidemic [42]. Indeed, the analysis of adult mosquito collections undertaken in eight localities in the region, including the main gold mining sites (Hafia, Wassangran, Ndebou, Matakossi, Bondala, Tenkoto, Velingra, and Bantako), revealed a high abundance of *Ae. furcifer* in these localities, except for Bantako, which exhibited a low abundance. While in Bantako, the species represented 7.9% of the mosquito fauna collected, in other localities the abundance ranged between 40.5% and 81.7% (Diallo, unpublished data). In addition, the univariate analysis showed that populations living close to the forest and to rivers (forest galleries) were significantly more exposed than the others (Table 1).

Although overall seroprevalence was low in the Bandafassi rural community, CHIKV seroprevalence was significantly higher in gold mining sites where the population density was relatively high, especially among male adults. In addition, during the outbreak in 2009–2010, those villages harboring such gold mining sites were most affected by CHIKV [43]. A similar pattern was also observed during the CHIKV outbreak in 2015, where confirmed cases in the Saraya district clustered in villages where the main gold mining sites were located (unpublished data). This suggests that traditional gold mining, by attracting thousands of indigenous and foreign populations to remote rural areas, particularly close to the forest galleries, may increase exposure of humans to CHIKV through spillover from the enzootic cycle. Moreover, environmental changes linked to human activity in sites with a high human concentration favor the development of domestic larval sites [43]. However, to better understand the distribution of CHIKV infection in the region, other potential risk factors that could have led to the higher risk, such as mosquito vector populations and the mobility of the gold miners, need to be investigated in the future. Although no chikungunya cases have been previously reported in the Salemata district, the seroprevalence rate was found to be high (>50%). This finding suggests either CHIKV circulation, with many mild or asymptomatic infections, which has been observed in only around 15% of infected individuals [13], or, more likely, it suggests a limited capacity of the surveillance system to detect cases. Limited surveillance in the Salemata district is potentially due to difficult access to health facilities. Indeed, the Salemata district is the most remote area of the region, without sentinel sites, in contrast to the Kedougou and Saraya districts, which have more robust surveillance.

The elevated exposure to CHIKV among human populations living in the rural Kedougou area suggests a high spillover risk from sylvatic into rural or domestic transmission cycles during amplification years. Particularly, gold mining sites that attract a large number of highly mobile individuals may act as hotspots for the emergence and dissemination of new CHIKV strains. Given the abundance of CHIKV vectors in the Kedougou region, the weakness of surveillance systems, and the mass human migrations, strengthening the surveillance system in the Kedougou region, including sensitization on environmental impact of gold mining activities is needed to prevent the establishment of a domestic, human-amplified CHIKV transmission cycle, and the potential global spread of newly introduced virus strains. Finally, due to the potential serologic cross-reactivity between CHIKV and alphaviruses, such as the o’nyong-nyong virus (ONNV), further studies are needed in order to assess the impact of these neglected arboviral diseases in the Kedougou region. However, the lack of ONNV, and other alphavirus detection in mosquito pools collected in the Kedougou region, suggests that the CHIKV seroprevalence that we measure does not represent cross-reactions.

## Figures and Tables

**Figure 1 viruses-12-00196-f001:**
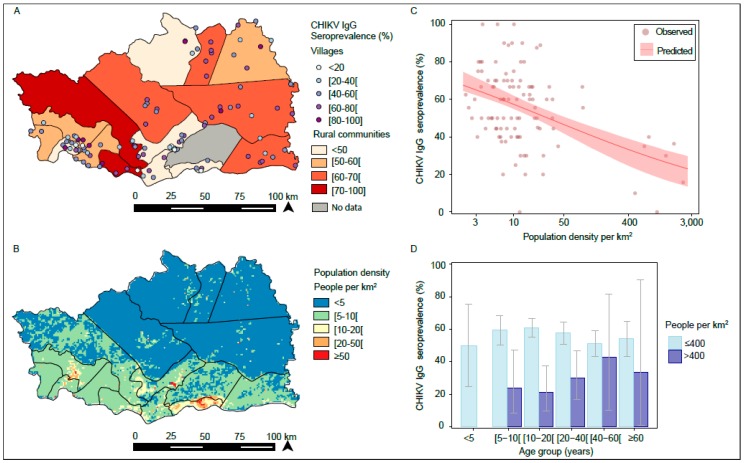
Spatial variation in chikungunya virus (CHIKV) IgG seroprevalence. (**A**) Seroprevalence by village and rural community. (**B**) Spatial variation in population density. (**C**) Observed and predicted seroprevalence by population density. The 95% confidence interval (CI) of the prediction was obtained by bootstrap (2000 iterations). (**D**) Age patterns by population density (≤400 vs. >400 people per km^2^).

**Figure 2 viruses-12-00196-f002:**
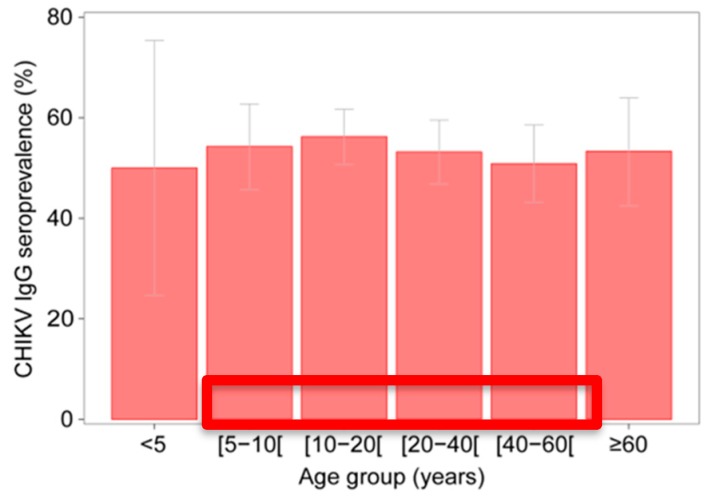
Seroprevalence by age group and exact binomial 95% confidence intervals.

**Table 1 viruses-12-00196-t001:** Univariable analysis of the association between CHIKV seroprevalence and environmental variables. The models were adjusted for clustering of individuals in villages and rural communities (random intercept).

Environmental Variables	OR (95%CI)	LRT *p*-Value	AIC
EVI_max (per 0.1 increase)	1.54 (1.16; 2.02)	0.002	1354
**EVI_mean (per 0.1 increase)**	**2.23 (1.44; 3.47)**	**<0.001**	**1351**
EVI_sd (per 0.01 increase)	1.14 (1.03; 1.27)	0.010	1357
**NDVI_max (per 0.1 increase)**	**1.90 (1.42; 2.57)**	**<0.001**	**1348**
NDVI_mean (per 0.1 increase)	1.85 (1.35; 2.52)	<0.001	1350
NDVI_sd (per 0.01 increase)	1.16 (1.03; 1.30)	0.012	1357
MIR_max (per 0.1 increase)	0.68 (0.35; 1.35)	0.258	1362
**MIR_mean (per 0.1 increase)**	**0.22 (0.10; 0.51)**	**<0.001**	**1352**
MIR_sd (per 0.01 increase)	1.02 (0.82; 1.28)	0.829	1363
**Distance to water bodies (km)**	**1.01 (1.00; 1.03)**	**0.048**	**1360**
Distance to rivers (km)	1.02 (0.99; 1.05)	0.202	1361
**Population density per km^2^ (log-transformed)**	**0.76 (0.69; 0.84)**	**<0.001**	**1340**
Slope (degree)	1.12 (0.98; 1.29)	0.089	1360
Altitude (meters)	1.00 (1.00; 1.00)	0.937	1363
Forest area (proportion, per 0.1 increase)	1.06 (0.99; 1.13)	0.081	1360
**Distance to forest (km)**	**0.86 (0.76; 0.98)**	**0.023**	**1358**
Accessibility (travel time to city per hour increase)	1.00 (1.00; 1.00)	0.985	1363
**Random Intercepts:**			
Village only	NA	0.003	1362
Rural community only	NA	0.094	1368
**Village and rural**	**NA**	**<0.001**	**1361**

**NDVI**: Normalized Difference Vegetation Index; **EVI**: Enhanced Vegetation Index; **MIR**: mid-infrared band; **OR**: Odds Ratio, **CI**: confidence interval; **LRT**: likelihood ratio test; **AIC**: Akaike information criterion.

**Table 2 viruses-12-00196-t002:** Multivariable analysis of the association between CHIKV seroprevalence and environmental variables. The models were adjusted for population density (log-scale) and clustering of individuals in villages (random intercept).

Environmental Variables	OR (95%CI)	LRT *p*-Value	AIC
Population density per km^2^ (log-transformed)	0.76 (0.69; 0.84)	0.008	1338
NDVI_max (per 0.1 increase)	1.17 (0.76; 1.81)	0.485	1340
Distance to forest (km)	0.97 (0.86; 1.10)	0.614	1340
Distance to water bodies (km)	1.00 (0.99; 1.01)	0.735	1340

**NDVI**: Normalized Difference Vegetation Index, **OR**: Odds Ratio, **CI**: confidence interval, **LRT**: likelihood ratio test, **AIC**: Akaike information criterion.

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
