# Peer review of "Changes in the Transmission Dynamic of Chikungunya Virus in Southeastern Senegal"

_viruses, 2020, doi:10.3390/v12020196_

Round 1
Reviewer 1 Report
This manuscript describes interesting results on the seroprevalence of CHIKV in the Kedougou region of Senegal in relation with the environement. The authors report that CHIKV seroprevalence was higher in population living close to the forest and was negatively correlated the density of the population. At the contrary, CHIKV seroprevalence was high in gold mining sites when population density was higher than 400 person/km2. They conclude that Aedes furcifer might play an important role in CHIKV transmission in forested areas, whereas in gold mining sites the presence of a high density of non native population and the development of domestic breading sites might increase the risk of human exposition in the region as well as the spread of new virus strains in other parts of Senegal due to the high mobility of the human population. There are some points that need to be clarified
In the abstract:
line 25
the region must be specified, Kedougou, localization,..
the ELISA test must be specific for CHIKV antibodies, therefore, the authors must write Samples were tested for specific anti-CHIKV...
In the introduction
line 55, Aedes albopictus mosquitoe introduction in Europe and USA is not due to the spill over of mosquitoes from Africa. This is an asiatic species and tyre importation explains most of the introduction of the mosquito.
In the Material and Methods:
The resolution of Figure 1 is not good. Colors can be confused especially red and pink.. It could also be interesting to present the density of the population on this map. We do not see the mining sites.
line 92: the authors should better describe the ELISA test used in this suty and explain in what it is specific to CHIKV. THere are other possible cross-reacting alphaviruses like O' Nyong Nyong for instance.
In the Results:
line 148/ Why is there such a bias in the sex of the testing individuals?
In the Discussion
line 195 the authors said that individuals living in remote areas with low population density were 1.24 times more exposed than those living in areas with high population density. Where does this figure come? Line 156 they said seroprevalence was 2 times higher in population living close to the forest. Are we talking about the same population. The authors should clarify this point.
Reviewer 2 Report
Major comments:
1. The authors have chosen to only conduct an ELISA to determine CHIKV seropositivity, but did not include any virus neutralization tests. As O'nyong-nyong virus (ONNV) has previously emerged in Senegal, it cannot be excluded that the virus is still circulating there. Serological cross-reactivity between alphaviruses is known and it can therefore not be excluded that a large proportion of the CHIKV-positive samples in the ELISA are actually positive for ONNV. As a result, it cannot be certain that the results are really demonstrating the transmission dynamics of CHIKV.
2. The conclusions of the authors linking population density and seropositivity may be valid, however, there are several important factors that may have influenced the observations (and therefore the scientific soundness) that have not been investigated or included in the study:
a) The authors speculate on the involvement of the Ae. furcifer vector in the rural areas and contributing to the high seropositivity there. However, can it be certain that this vector is not also active at the gold mining sites? What vectors are active at the gold mining sites? And what are the differences in abundance and behavior of mosquitoes between the <400 and >400 km2 population areas? These may all influence the observed differences in seroprevalence between the two area types. Has any mosquito trapping been conducted?
b) The number of individuals tested per village and rural community have not been indicated. In particular it would be relevant to know how many individuals were tested in the <400 km2 area compared to the >400 km2. If only a very small number of individuals has been tested in the rural areas compared to the more populated areas it may skew the results.
c) As mentioned in the discussion, the male working class is overrepresented in the population structure of the mining towns. This would not make it surprising that the seroprevalence in this group is the highest. However, a representative table indicating the numbers per age category and gender would be insightful.
d) Also, how constant is the population in the mining towns? Often the workers are highly mobile, and continuously move to new sites. This may also affect the observed seroprevalence. Was any such information recorded as part of the serosurveys? The authors also mention that the gold mining sites have attracted indigenous and foreign populations to these areas. Has it been reported in the serosurvey where the sampled persons have originated or travelled from? This could also heavily influence the seropositivity reported in the gold mining sites.
3. A map additional to Fig 3A and B showing the spatial variation of forest cover would be useful, as it will allow the visual comparison of population density and forest cover.
Minor comments:
Line 2: The word "Changes" would suggest that serosurverys were conducted over a long period of time and that a shift in the observations occurred after x amount of time. Also, consider making dynamic plural.
Line 3: Remove the comma.
Line 24: Decapitalize the O in One.
Line 25: Include a sentence that mentions that serum samples were taken from individuals. State in which region samples were taken.
Line 28: Was 54% positive in the entire sampled region? If so, specifiy this behind "individuals.."
Line 33-34: This sentence is too presumptive. It assumes that the vector in the gold mining sites is different, but this information is not available or has not been presented by the authors.
Line 52: Indicate in which year the outbreak in La Reunion occurred.
Line 54: "elsewhere in Europe" is too broad, specify the countries. For USA, specify which parts of the US.
Lines 70-74: The authors state that limited information is available on the impact of the virus on human health, but this study does not investigate anything in terms of human health. Furthermore, the virus does not interact with environmental conditions but the vector does. Instead, state more clearly the objectives of the study, which seems mostly to be to determine the correlation between CHIKV seroprevalence, forest cover and population density.
Line 89: The word "unless" does not fit in the sentence at all and should be changed to another word or removed.
Line 91 & 92: "were" should be "was"
Lines 98-99: Like for MIR, consider adding a sentence that states what the NDVI and EVI actually discriminate.
Line 99: Delete the extra space.
Figure 1 remained of too low resolution.
Line 110: Add another space between "to" and"59".
Line 117: Sentence should be: "considering that age groups reflect..."
Line 156: Consider deleting significantly and instead stating that seroprevalence was two times higher, with the P-value behind it. Also, at the end of line 157, mention what it was higher in comparison to.
Line 162: Remove the full-stop behind "(Figure 3B)"
Line 173: In contrary = On the contrary.
Line 185: Boostrap should probably be bootstrap.
The caption for Figure 3 could provide more information. For example, for (D), it also shows seroprevalence but this is not mentioned.
Line 202: "were" should be "was"
Line 204: habor should be harboring
Line 210: There is an extra space in front of "Although"
Line 214: Extra space in front of Indeed. Also consider changing the sentence that starts with Indeed to "Indeed the lack of surveillance in this area is potentially due to difficult access to health facilities in this district, which is the remotest area of the region with absent sentinel sites, in contrary to Kedougou and Saraya districts."
Line 222: Consider changing part to: "...the weakness of surveillance systems and the mass human migrations, it is urgent to strengthen..."
Line 224: I do not think it is possible that strengthening a surveillance system can prevent the establishment of a domestic CHIKV transmission cycle. This statement is too bold.
Author Response
Please attachments

Reviewer 3 Report
Summary
Changes in the transmission dynamic of Chikungunya virus in South Eastern, Senegal by Sow et al., describes the testing of 120 persons (villagers) in Senegal by random selection. Plasma samples were then obtained and screened for the presence of CHIKV IgG antibodies by ELISA. Associations of CHIKV seroprevalence were assessed using regression analysis and the spatial correlation of village seroprevalence.
Major findings 1) that 54% of individuals across all tested age ranges were tested positive for CHIKV-specific IgG. CHIKV. CHIKV seroprevalence was higher near forested areas and peaked unusually near mine facilities in the region.
Overall, this brief report is interesting and potentially important to the field. There are some concerns that should be addressed that include some language clarifications, better quality figures.
Major:
The manuscript is generally well-written and interesting to read. There are several English and grammatical errors that should be corrected but these are not serious and it was assumed that English is likely one of 3 or four languages spoken by the key author. There may also be some minor formatting issues. The main figures #1-3 are difficult to read and are very low quality when increased in size. Higher quality images should be used. The high seroprevalence of CHIKV Ab in gold mine regions was only briefly evaluated in the discussion- this is a potentially important finding, but potential confounders were not discussed in detail.
Minor:
1. The manuscript requires a thorough read for minor language errors.
Round 2
Reviewer 2 Report
In my opinion, one of the major flaws in this article is that the gold standard for verification, the virus neutralization assay, was not used to confirm that the ELISA-positive results were indeed CHIKV, instead of ONNV. The authors included a comment in the discussion to suggest that further research should be conducted to understand the burden of ONNV in this region in the future. I think this is too simplistic, and this current study should be used as an opportunity already to investigate this burden. I understand that there may be limited resources to conduct VNTs, but then this manuscript should go to a journal with a lower impact factor.